# Establishment and Natural Regeneration of Native Trees in Agroforestry Systems in the Paraguayan Atlantic Forest

**Amado Insfrán Ortiz** [1,2,*] **, José María Rey Benayas** [2] **and Luis Cayuela** [3]

1    Faculty of Agrarian Sciences, National University of Asunción, San Lorenzo University Campus, km 10, Asunción 1618, Paraguay
2    Forest Ecology and Restoration (FORECO) Group, Life Sciences Department, University of Alcalá, 28871 Alcalá de Henares, Madrid, Spain
3    Department of Biology and Geology, Physics and Inorganic Chemistry, Rey Juan Carlos University, c/Tulipán s/n, 28933 Móstoles, Madrid, Spain
*    Correspondence: amado.insfran@agr.una.py

**Abstract:** The establishment of planted trees and the natural regeneration of trees in agroecosystems is challenging. This study evaluated the establishment and natural regeneration of the following six native tree species in two agricultural systems in the Atlantic Forest in Paraguay: *Cedrela fissilis* Vell., *Cordia trichotoma* (Vell) Arráb. ex Steud., *Handroanthus albus* (Cham.) Mattos, *Handroanthus impetiginosus* (Mart. ex DC.) Mattos, *Peltophorum dubium* (Sprengel) Taubert, and *Cordia americana* (L.) Gottschling and J.S.Mill. At the study site in Caaguazú, 18 plots of 100 m$^2$ each were established in 2 agronomic systems (conventional or agroecological) featuring 3 plantation types (pathsides, agricultural field edges, and islets). Trees were planted at this site in spring 2010 at a density of 1800 individuals ha$^{-1}$, and the site was monitored for six years. At the study site in Itapúa, 30 plots of 50 m$^2$ each were established in three agronomic systems (conventional, traditional, or agroecological). Trees were planted at this site in spring 2012 at a density of 1600 individuals ha$^{-1}$, and the site was monitored for four years. Survival and relative growth rates of the planted species and natural regeneration were analyzed using generalized linear mixed models that considered species, agronomic system, and plantation type as fixed factors, and time and plot as random factors. At both sites, survival varied among species. Here, *C. fissilis* showed lower survival and *C. trichotoma* higher growth than the other species. Naturally regenerated species were *C. trichotoma*, *H. albus*, and *P. dubium*. The agronomic system and species affected growth and natural regeneration at both locations. Plantation type affected survival and growth in Caaguazú only. We conclude that species contributes more than agronomic system or plantation type to determining the survival, relative growth rate, and natural regeneration in agroforestry systems in the Paraguayan Atlantic Forest.

**Keywords:** agroecosystem; agroecology; growth; recruitment; subtropical; survival





## 1. Introduction

Reduction in forest areas, mainly because of the expansion of agriculture [1,2], causes the loss of biodiversity and habitats and affects the provision of ecosystem services [3–5]. In tropical and subtropical countries, agricultural expansion is responsible for 73% of deforestation [6] and is a leading cause of species decline, including 74% of threatened bird species [7]. This has reduced the provision of ecosystem services, particularly those with regulatory functions, such as soil retention, runoff control, and nutrient cycles [8]. The Atlantic Forest in Paraguay has been severely affected by large-scale habitat loss and fragmentation [9].

One way to counteract the harmful effects of agriculture is establishing and maintaining agroforestry systems, where wildlife-friendly farming that allows more sustainable agricultural land use is practiced [10,11]. Agroforestry systems deliberately combine trees and shrubs with some crops [12], leading to levels of biodiversity and ecosystem

services intermediate between those in agricultural areas without trees and those of native forests [13,14]. Thus, the combination of native tree species with agricultural crops is a valuable strategy that integrates agricultural production and biodiversity conservation [3,11,15,16]. Unfortunately, this strategy still faces economic obstacles and so has not been widely applied [11,17].

Trees can be planted in agricultural landscapes as linear elements, such as living fences, on the edges of agricultural fields, pathsides, and banks [10,18] and as islets, thereby minimizing competition for agricultural space [19]. Both strategies provide a variety of environmental goods and services [20], such as the maintenance and improvement of biodiversity [13], including pollinators and enemies of natural pests [21], the regulation of runoff and sediment and nutrient retention [1], water infiltration [22], production of edaphic organic matter [23,24], microclimate regulation [25], landscape connectivity [18], scenic beauty [26], and crop productivity [1]. In addition, agroforestry systems can provide benefits that help reduce pressure on the remaining forests [27]. For example, planted trees can export seeds and nucleate the growth of other shrub and tree species, constituting a system straddling between a naturally regenerated forest and a forest plantation. They allow large-scale passive reforestation, where vegetation and soil can sequester carbon [19,28].

The establishment of native trees in tropical and subtropical agroecosystems, whether in plantations or as part of natural regeneration, depends mainly on the species, management after forest plantation, and type of agroecosystem [29,30]. Such species are often planted in sections of agricultural fields or in cattle grazing areas [31,32]. In agricultural landscapes, native trees can be established along pathsides, the edges of agricultural fields, and in islets [33], which are forms of nucleation [34]. However, the factors affecting the success of these efforts at plantation and natural regeneration sites are unclear.

Thus, the objective of the present study was to evaluate the establishment (survival and growth) and natural regeneration of native tree species planted in linear form (pathsides and field edges) or islets (Figure S1) in conventional, traditional, or agroecological agricultural systems in the Atlantic Forest in Paraguay. The differences between these systems are indicated in Table S1. For this purpose, experimental plantations were monitored for 4–6 years. Our hypotheses were that tree survival, growth, and regeneration would depend on the following: (1) the species planted; (2) the agronomic system, with agroecological systems being the most favorable for the establishment and natural regeneration of planted trees because of the high soil quality and absence of pesticides; and (3) the type of plantation, with islets providing the best establishment and natural regeneration because they are less exposed to sun, livestock, and wind. The results of this work may help guide the transformation and conversion of conventional treeless agroecosystems into more sustainable agroforestry systems in the Neotropics.

## 2. Materials and Methods

### 2.1. Study Area

The study was conducted in the municipalities of Repatriación (Department of Caaguazú) and of Carlos Antonio López (Department of Itapúa) in the Atlantic Forest of Paraguay (Figure 1).

The climate in this area is characteristic of the semi-deciduous subtropical Atlantic Forest, with frequent rains totaling 1300–1800 mm/year and an average annual temperature of 22 °C, with a minimum of 0 °C (winter) and a maximum of 42 °C (summer) [36]. The soil in the two areas differs. In the plantation area in Caaguazú, the soil is of sandstone origin and belongs to the Ultisol order and arenic rhodic paleudult sub-group (classes III and IV) in the sand-clay textural subdivision containing 20%–30% of clay and showing strong water erosion and high degradation [37]. In the plantation area in Itapúa, the soil is of basaltic origin and belongs to the Ultisol order and rhodic paleudult subgroup (class II) in the fine clay textural subdivision containing 30%–40% of clay and showing moderate water erosion and medium-level degradation [37]. The main economic activities in the two areas are agriculture, mostly the cultivation of cotton, corn, sugarcane, beans, yerba

mate, cassava, wheat, and soybean, with a predominance of monocultures in conventional systems; and the rearing of cattle, pigs, and poultry [38].

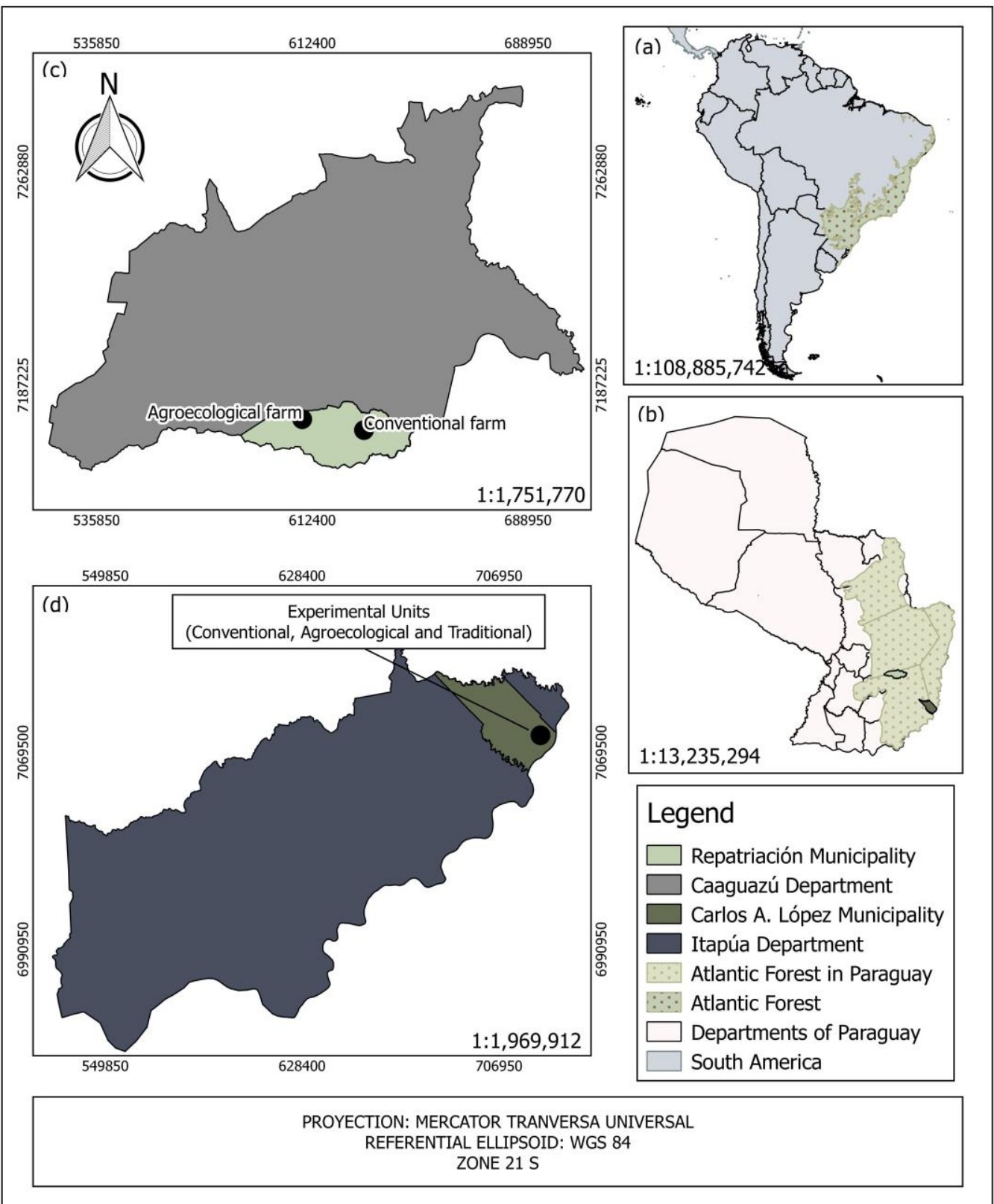

**Figure 1.** Identification of the study area (**a**) in the Neotropics and (**b**) in the eastern region of Paraguay, and location of the experimental plantations in (**c**) Caaguazú and (**d**) Itapúa. Panels a and b show the distribution area of the Atlantic Forest according to [35].

The Atlantic Forest, one of the most biodiverse biomes in the world, is also one of the most threatened [39]. It occupies an estimated area of 8,500,000 ha in Paraguay [40], accounting for 55% of the country's eastern region [36]. Most of this native forest was intact in 1940 [41], and the intact proportion fell to 73.4% by 1973 and to 24.9% by 2000 [42], and it stands now between 10 and 13% [43]. The main drivers of forest loss and degradation in this region are large-scale agriculture [44], mainly the cultivation of soybean [45], small landowner settlements, and the extraction of high-value timber species, such as *Amburana cearensis* (Allemão) SCSm., *Cedrela fissilis*, and *Myrocarpus frondosus* Fr. Allem [46].

## 2.2. Experimental Description

The study included two departments (Caaguazú and Itapúa), three agronomic systems (conventional, traditional, and agroecological), and three plantation types (linear on pathsides, edges of agricultural fields, and islets; Table 1).

**Table 1.** Experimental plantations established at the study sites.

| Location | Agronomic System | Plantation Type | No. Plots | Plot Size (m²) | Tree Density (Individuals ha⁻¹) | Monitoring Period (Years) * |
|---|---|---|---|---|---|---|
| **Caaguazú** | Conventional | Pathside | 3 | 100 | 1800 | 2010–2016 |
| | | Field edge | 3 | 100 | 1800 | 2010–2016 |
| | | Islet | 3 | 100 | 1800 | 2010–2016 |
| | Agroecological | Pathside | 3 | 100 | 1800 | 2010–2016 |
| | | Field edge | 3 | 100 | 1800 | 2010–2016 |
| | | Islet | 3 | 100 | 1800 | 2010–2016 |
| | | Total | 18 | | | |
| **Itapúa** | Conventional | Islet | 10 | 50 | 1600 | 2012–2016 |
| | Traditional | Islet | 10 | 50 | 1600 | 2012–2016 |
| | Agroecological | Islet | 10 | 50 | 1600 | 2012–2016 |
| | | Total | 30 | | | |

* Years monitored are as follows: 2010, 2012, 2014, and 2016. The monitoring of survival, growth, and natural regeneration was carried out from the first year of establishment of the plots (month of December).

The conventional agronomic system primarily includes large plots (>50 ha) where high-yield seed varieties [47] are cultivated using large amounts of agrochemicals [48,49] and mechanization; the production is mainly for export [50]. The traditional system, in contrast, is practiced by small farmers (plots < 20 ha) [49], who in Paraguay are mainly family farmers who cultivate plots < 10 ha [51]. Traditional agriculture is based on technological, cultural, and local knowledge practices, diversified crops [52], and minimal mechanization [53]. The agroecological system is practiced by small farmers [54] and is based on farmers' knowledge, ecological principles, cultivated and uncultivated biodiversity [55], resistant and resilient systems, energy efficiency, and social justice [56]. In this system, pest management, nutrient cycles, biomass recycling, biological interactions, and biodiversity synergies are leveraged to promote ecosystem services, agricultural production, and biodiversity conservation [57,58].

In Caaguazú, tree plantations were established on two farms. The two farms were similar in terms of soil conditions, terrain, elevation, and climate, in order to minimize potential confounding effects between farm and agroecological system. However, the farm management is different. One was managed according to conventional practices (coordinates 25°34′37.30″ S and 55°45′01.53″ W, average altitude of 270 m above sea level), and the other according to agroecological practices (coordinates 25°33′11.25″ S and 55°55′38.02″ W, average altitude of 325 m above sea level), Table S1. The three types of tree plantation were established in each farm/agronomic system, including three plots of 100 m² each for each plantation type. Plot dimensions were 10 m × 10 m in the case of pathsides and islets, or 2.5 m × 40 m in the case of agricultural field edges (Table 1).

Six native tree species were planted in October 2010, as follows: *H. albus*, *H. impetiginosus*, *P. dubium*, *C. fissilis*, *C. trichotoma* (a tree species valued for wood furniture), and *C. americana* (valued for rural buildings, firewood, and poles). All these species occupy the arboreal stratum (20–30 m high) of the Atlantic Forest in Paraguay. *Ilex paraguariensis* Saint Hilaire (yerba mate) was also planted and shares the space as an economically important annual cut crop, but it was excluded from this study. The planting density of the native species was 1800 individuals $ha^{-1}$ (18 individuals/plot), and that of *I. paraguariensis* was 500 individuals $ha^{-1}$ (5 individuals/plot), with an average distance of 2.5 m between plants (Figures S1 and S2). The individuals were planted at a depth of 20 cm and irrigated at 30 L $m^{-2}$ during the first year after planting to help them withstand the first summer drought.

In Itapúa, plantations were established on a farm that was managed traditionally (coordinates 26°24′56.87″ S and 54°45′33.57″ W, average altitude of 209 m above sea level). Thirty experimental plantation plots were established only as islets in the three agronomic management systems. Each system included 10 plots of 2.5 m × 20 m (Table 1). Plantations were established in October 2012 using five native tree species, i.e., the same as in Caaguazú except for *C. americana*, which was unavailable in nurseries. The species were planted at a density of 1600 individuals $ha^{-1}$ (8 individuals/plot), together with *I. paraguariensis* (2600 individuals $ha^{-1}$, 13 individuals/plot). The plantations were established after a precipitation of 40 L $m^{-2}$. The spatial arrangement was "staggered", with 2 m between plants in the same row and 2.5 m between rows. The planting depth was 15 cm (Figure S3).

The differences between the two sites in planting density, planting depth, and watering reflected the fact that the sandy soil in Caaguazú was looser and retained less water than the basaltic soil in Itapúa. Plantation management also differed between the two locations in terms of soil tillage, fertilizer, *I. paraguarienses* crop density, and annual practices to control weeds and harmful insects (Table S1). The initial average heights of the planted seedlings are reported in Table S2.

### 2.3. Monitoring the Establishment and Natural Regeneration of Tree Species

Survival of planted individuals was monitored in spring (October and November) in the years 2012, 2014, and 2016 in Caaguazú, and the years 2014 and 2016 in Itapúa. The number of surviving or dead individuals and the apparent causes of mortality were recorded at all plantation plots. Simultaneously, the numbers of individuals of the different species that were established through natural regeneration were recorded. Tree height ($x$) was measured to estimate the relative growth rate in October and November, which was calculated as follows:

[$\ln(x)$ at the last measurement $- \ln(x)$ at the first measurement]/total no. of days.

### 2.4. Data Analysis

We applied generalized linear mixed models to study the effect of the species planted, the agronomic system, and the type of plantation on survival, growth, and natural regeneration of seedlings over time at the two locations. In Itapúa, as there was only one form of plantation (islet), this effect was not studied. We used different error distributions for each response variable, as follows: binomial for survival (proportion data), gamma for growth (continuous data), and Poisson for natural regeneration (count data). The complete model incorporated the species, agronomic system, and plantation type as fixed factors, and the sampling year as a covariate, including all possible pairwise interactions among these predictor variables and the triple interaction among species, agronomic system, and plantation type. Plot was included as a random factor.

All possible sub-models were compared using the Akaike information criterion corrected for small sample sizes (AICc), and models with $\Delta$AICc $\leq 2$ with respect to the best model were selected [59]. In cases with more than one 'best model', the most complex model was selected. The residuals of the best model(s) were explored using a simulation-based approach [60]. Multiple post-hoc comparisons were conducted using Tukey tests,

with a significance level of 0.05, to determine the effects of different combinations of agronomic systems and plantation types on each of the response variables for each of the study species.

All analyses were performed with R [61] using the packages 'lme4' [62], 'MuMIn' [63], 'multcomp' [64], and 'DHARMA' [65].

## 3. Results

### 3.1. Survival of Planted Species

In Caaguazú, comparison of models with $\Delta$AICc $\leq$ 2 indicated that the planted species, agronomic system, and plantation type influenced survival probability (Tables 2 and S3a). At six years after planting, the studied species showed different average survival rates independently of other factors; *C. fissilis* showed the lowest survival rate (33.3 $\pm$ 34.3%) and *C. americana* the highest (91.7 $\pm$ 19.2%; Table S4a). Plantations in the agroecological system had higher survival rates (69.1 $\pm$ 14.8%) than in the conventional system (57.4 $\pm$ 24.3%). The higher survival in agroecological systems depended on (1) the type of plantation, with higher survival on pathsides and agricultural field edges, and (2) the species, with agroecological systems favoring the survival of *C. fissilis*, *P. dubium*, and *H. impetiginosus* more than the survival of *H. albus*, *C. trichotoma*, and *C. americana*. The effect of the agronomic system was not observed for the species as a whole (Figure 2).

**Table 2.** Plausible models for the survival of planted species in the two agroforestry experiments.

| Site | Model | (Intercept) | AICc | Delta | Weight | $R^2m$ | $R^2c$ |
|------|-------|-------------|------|-------|--------|--------|--------|
| Caaguazú | 512 | 3,590,128 | 959.8110 | 0.0000000 | 269,608.5 | 0.8687040 | 0.8748712 |
| | 1024 | 3,750,841 | 960.4830 | 0.6720045 | 192,668.1 | 0.8671280 | 0.8733760 |
| | 384 | 3,656,629 | 961.1081 | 1.2971396 | 140,949.4 | 0.8633484 | 0.8693383 |
| Itapúa | 24 | 0.5699077 | 556.5128 | 0.000000 | 863,392.6 | 0.2894882 | 0.2924227 |

Abbreviations are as follows: AICc, Akaike information criterion corrected for $\Delta$AICc $\leq$ 2; $R^2$c, conditional $R^2$ indicating the variability explained by the fixed and random effects; $R^2$m, marginal $R^2$ that considers only the variability explained by the fixed effects.

In Itapúa, the comparison of models with $\Delta$AICc $\leq$ 2 showed a significant effect of species and agronomic system on probability of survival (Tables 2 and S3a). The five species analyzed had different average survival rates (Figure 2 and Table S4a). Again, *C. fissilis* had the lowest survival rate (33.3 $\pm$ 47.9%), while *C. trichotoma* had the highest survival (75.00 $\pm$ 36.55%). The agronomic system did not significantly affect the survival of *C. fissilis* or *H. albus*, in contrast to the rest of the species (Figure 2 and Table S4a).

### 3.2. Growth of Planted Species

Comparison of models for Caaguazú indicated that the species, agronomic system, and plantation type influenced the growth rate of individuals. In Itapúa, however, only the species had a significant influence (Table 3). The species differed in their relative growth rates after 2212 days since planting in Caaguazú and after 1425 days in Itapúa (Figure 3a,b and Table S4b). At both sites, *C. trichotoma* showed higher relative growth than the other species (Figure 3). The agronomic system affected the relative growth rate in Caaguazú, being higher in the conventional system (0.00144 $\pm$ 0.00066 m day$^{-1}$) than in the agroecological system (0.00139 $\pm$ 0.000234 m day$^{-1}$, $p \leq$ 0.01), but this was not observed in Itapúa (conventional at 0.00211 $\pm$ 0.000108 m day$^{-1}$, agroecological at 0.00214 $\pm$ 0.000096 m day$^{-1}$, and traditional at 0.00213 $\pm$ 0.000159 m day$^{-1}$) (Figure 3a,b and Table S4b). The relative growth rates in Caaguazú differed across the different plantation types in the following order (Figure 3c and Table S4b): field edges (0.00149 $\pm$ 0.000203 m day$^{-1}$) > pathsides (0.00147 $\pm$ 0.000257 m day$^{-1}$) > islets (0.00136 $\pm$ 0.000265 m day$^{-1}$).

**Table 3.** Plausible models for the growth of the planted species in the two agroforestry experiments.

| Site | Model | (Intercept) | AICc | Delta | Weight | $R^2m$ | $R^2c$ |
|------|-------|-------------|------|-------|--------|--------|--------|
| Caaguazú | 40 | 6,978,656 | 298.5042 | 0.000000 | 946,708.8 | 0.8671280 | 0.8733760 |
| Itapúa | 2 | 0.2708572 | 638.4951 | 0.000000 | 888,152.8 | 0.2894882 | 0.2924227 |

Abbreviations are as follows: AICc, Akaike information criterion corrected for a $\Delta AICc \leq 2$; $R^2c$: conditional $R^2$ indicating the variability explained by the fixed and random effects; $R^2m$, marginal $R^2$ that considers only the variability explained by the fixed effects.

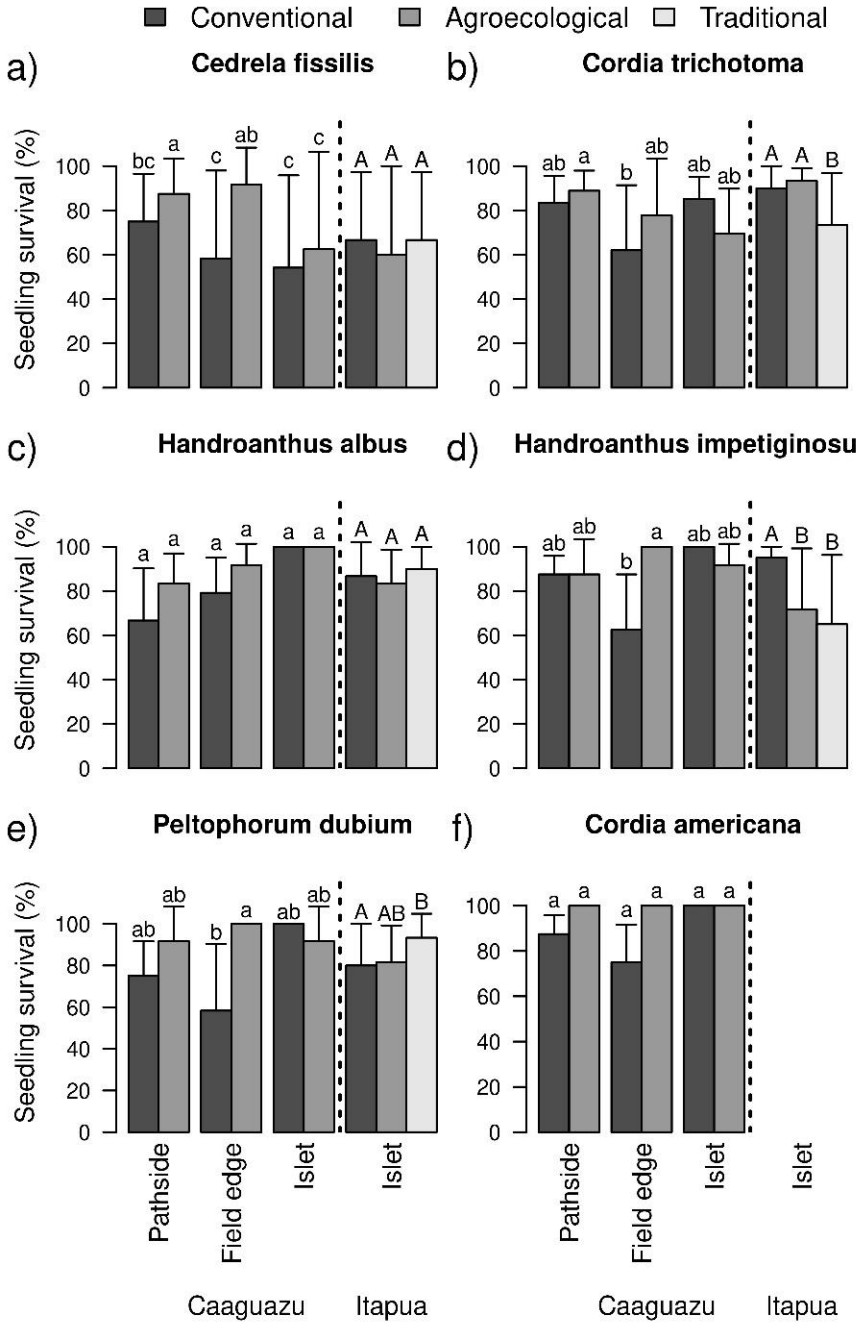

**Figure 2.** Survival of the planted species in different agronomic systems and plantation types. The time interval in Caaguazú and in Itapúa was from 2010 to 2016 and from 2012 to 2016, respectively. The subfigures (**a–f**) are the species. The letters "a", "b", "c", "ab", "bc", "A", "B" and "AB", are statistical differences and similarities between agricultural systems.

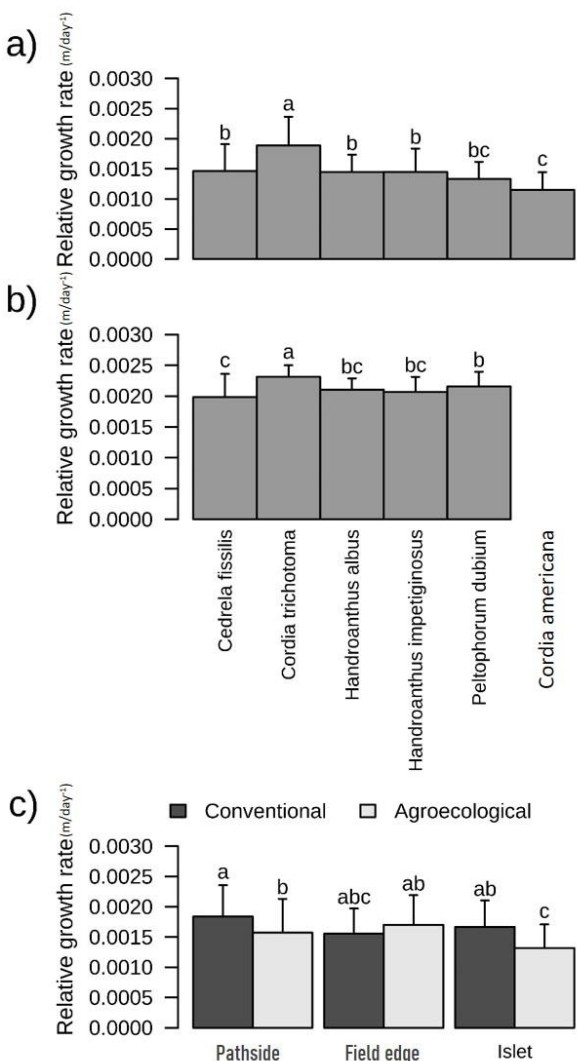

**Figure 3.** Relative growth rate (m day$^{-1}$) of native species planted in (**a**) Caaguazú and (**b**) Itapúa. (**c**) Comparison of relative growth rates for different agronomic systems and plantation types in Caaguazú. The letters "a", "b", "c", "ab", "abc" and "bc", are statistical differences and similarities be-tween species and be-tween agricultural systems.

### 3.3. Natural Regeneration

Comparison of models indicated a significant effect of species and agronomic system on the natural regeneration of planted species at both locations (Table 4).

**Table 4.** Plausible models for the regeneration of planted species in the two agroforestry experiments.

| Site | Model | (Intercept) | AICc | Delta | Weight |
|------|-------|-------------|------|-------|--------|
| Caaguazú | 22 | −193,025,850 | 159.8007 | 0.000000 | 676,183.7 |
|  | 24 | −200,038,011 | 161.6283 | 1.827547 | 271,155.1 |
| Itapúa | 4 | −18.560 | 130.8007 | 0.000000 | 676,183.7 |

Abbreviations are as follows: AICc, Akaike information criterion corrected for $\Delta$AICc $\leq$ 2.

Of the six planted species, we observed only non-planted, naturally regenerated individuals of *C. trichotoma*, *H. albus*, and *P. dubium* (Figure 4 and Table S4c). In addition, one individual of a non-planted species (*Cecropia pachystachya* Trécul.) was identified in Caaguazú. At that site, the number of naturally regenerated *H. albus* individuals was higher in the conventional system (3.00 ± 3.43 individuals/plot) than in the agroecological

system (0.11 ± 0.33 individuals/plot; Figure 4b and Table S4c). In Itapúa, the natural regeneration of the three mentioned species differed between agronomic systems, being higher in the agroecological system (0.50 ± 0.80 individuals/plot) than in the conventional (0.17 ± 0.38 individuals/plot) and traditional (0.10 ± 0.31 individuals/plot) systems (Figure 4 and Table S4c). The agronomic system and plantation type did not affect the natural regeneration of *C. fissilis*, *H. impetiginosus*, or *C. americana* in Itapúa (Table S4c).

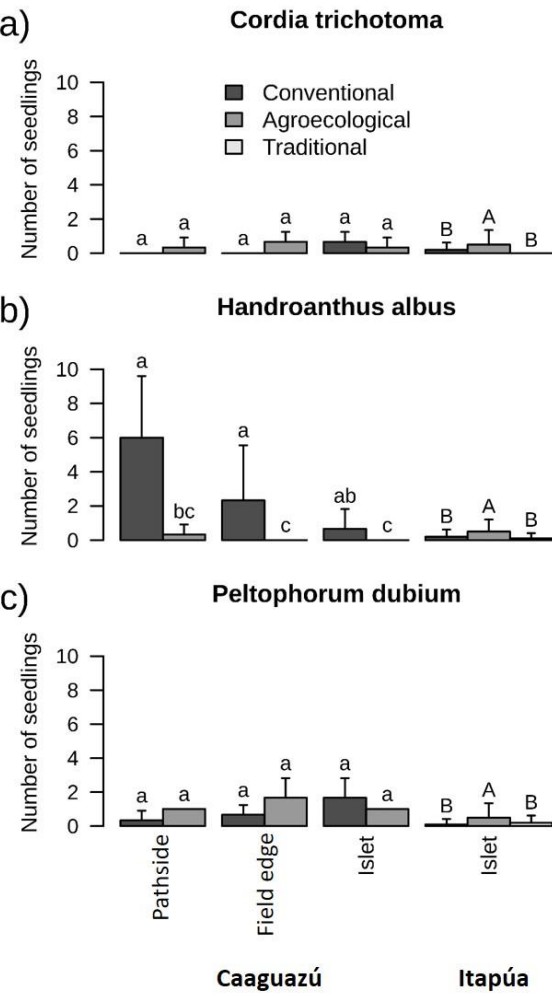

**Figure 4.** Number of naturally regenerated individuals of (**a**) *Cordia trichotoma*, (**b**) *Handroanthus albus*, and (**c**) *Peltophorum dubium* in Caaguazú and Itapúa. The letters "a", "b", "c", "ab", "bc", "A" and "B", are statistical differences and similarities between types of plantations.

## 4. Discussion

The establishment of native trees and natural regeneration in *I. paraguariensis* crops in the Atlantic Forest has rarely been investigated in a systematic fashion. This study analyzed the establishment (survival and growth) and natural regeneration of six species planted in different agronomic systems and plantation types in the Atlantic Forest region in Paraguay. In general, survival, growth, and natural regeneration at the two study sites depended on the planted species (H1). However, the differences among the agronomic systems (H2) and plantation types depended on the study site (H3).

### 4.1. Establishment and Natural Regeneration Depend on the Species

Our results indicated that the species systematically affects survival, growth, and natural regeneration at the two study sites. Survival averaged 65.9% after six years of monitoring in Caaguazú and after four years in Itapúa; in some cases, it was higher than that reported for other species of the same genus. In Venezuela, for example, *Cordia thaisiana*,

*Cedrela odorata*, and *Handroanthus rosea* showed survival rates of 56%–81% at 18 months after planting in the subtropics, while two species of *Cedrela* showed survival rates of 74 and 81% at 24 months after planting in the humid tropics [30]. Other studies reported survival rates of 60% for *Cordia alliodora* after six years of planting [66] and 50% for *C. odorata* [67]. In an agroforestry system in Rwanda, the survival rate for *Cedrela* spp. and *C. trichotoma* was 60%, and the mean rate across the 14 studied tree species there was 51% [29]. Additionally, *C. trichotoma* survival reached 70% in our study, consistent with a previous report that its survival rate was higher than that of other species [68].

In our study, a crucial determinant of the survival of planted trees was browsing by cattle from neighboring farms, which affected 94% of the experimental plots. This highlights that proper management of livestock is essential in agroforestry [29]. For example, in Zambia, owners who controlled their livestock achieved survival rates of 68% in agroforestry systems, compared to only 32% on farms without this control [69]. Therefore, raising fences to exclude livestock may significantly increase tree survival rates [70]. Even so, the survival rates in our studies were higher than those reported in previous work.

We attribute the low survival of *C. fissilis* not only to cattle browsing but also to the vulnerability of this species to leafcutter ants, whose presence can be observed from the first year of planting [71,72], to the mahogany shoot borer *Hypsipyla grandella* (Zeller), which commonly infests this species [73,74], and/or to diseases, such as canker fungus (*Botryosphaeriaceous fungi*) [75]. In agreement with previous work [76], our field observations indicated greater attacks of leafcutter ants on *C. fissilis* than on *P. dubium* and *H. impetiginosus*, and no attacks on *C. trichotoma* or *H. albus*, which survived longer than *C. fissilis*. Agroforestry systems, by promoting greater biodiversity than simple crops, can promote the presence of natural enemies of leafcutter ants [77].

The relative growth rate in our study, which averaged 0.00176 m/day$^{-1}$ across all species, varied substantially from one species to another. The higher growth rate of *C. trichotoma* compared to the other species is consistent with previous observations [78], and in part it may reflect that the tree participates in mutualism with *Hymenoptera*, as it was found between *Cordia eliodora* and Astecas ants [79]. Conversely, the low growth rate of *C. fissilis* can be attributed to attacks by *Atta sexdens* and *H. grandella*, as reported in several studies [80,81]. The higher growth rate in the conventional agronomic system in Caaguazú may be attributable to soil characteristics (Table S5), such as texture, structure, soil moisture, and fertility [82], which favor moisture retention, which in turn promotes water consumption and plant growth [83] or forms of eradication of competitive weeds [84].

In our study, only three species showed natural regeneration. The seedlings of the pioneers *C. trichotoma* and *P. dubium* [85,86] came from seeds of adult congeners located near the planting plots. The *P. dubium* species grows fast [85] and its serotinous character leads to high seed germination [87,88]. Unlike these two species, *H. albus* flowers at an early age (in year four), allowing the appearance of seedlings in situ, which is important because it is a threatened species in Paraguay. Unfortunately, *C. fissilis,* a threatened species in Appendix II of the report in [89], did not show regeneration in the experimental plots, as observed in Costa Rica [90]. This is attributable to the long time needed for *C. fissilis* to produce seeds [70] and the absence of nearby mother plants that could export seeds to agricultural plots [91].

Intensive farms generally have very few or no mature trees [92] and in agroforestry systems, farmers perform annual plantings that make natural regeneration difficult [93,94]. However, in agroecosystems receiving minimal human intervention, such as natural grasslands or conservation areas, many tree species can reproduce naturally [95,96].

*4.2. Effects of the Agronomic System and Plantation Types on Establishment and Natural Regeneration*

We tested the hypothesis that agronomic systems would affect the establishment and natural regeneration of the species, and that agroecological systems would facilitate these outcomes. This hypothesis was partially confirmed, since growth was similar in different

agronomic systems, and natural regeneration was greater in agroecological plantations only in Itapúa. Higher survival in agroecological than conventional systems is consistent with previous work [93] on tropical ecosystems in Costa Rica. In our study, the higher survival can be attributed to two factors. First, the initial soil quality was better in the agroecological than conventional system; pH was close to neutral, the concentration of exchangeable aluminum was lower, and there was less compaction and greater depth. This better edaphic quality in agroecological systems has been repeatedly demonstrated [97,98]. Second, in the conventional system, the use of glyphosate on our experimental plots or contiguous plots may have weakened the ability of tree seedlings to resist environmental stress and fungal diseases [99], and these effects can occur up to 20 m from the point of application, since the wind disperses the glyphosate [100]. In contrast, growth was slightly higher in the conventional than agroecological system in Caaguazú, which may be due to the use of insecticides to control leafcutter ants in the first system [30,67], and to the greater organic matter content and better texture of the conventional soil [5] (initial soil analysis in the experimental plots is shown in Table S5).

We investigated whether the establishment and natural regeneration would be more significant in islets than in pathsides and agricultural field edges, since these linear elements are more exposed to sun, livestock, and wind. This hypothesis was partially confirmed, since the type of plantation affected both survival and growth in Caaguazú, but it did not affect natural regeneration at either site. These results could be attributed to the following two factors: first, the scarcity of mother plants in the vicinity of the plots as sources of propagules [19]; second, the presence of livestock in the experimental plots due to a lack of local management [29].

## 5. Conclusions

The results of this study will help to implement agroforestry systems in the subtropics of Latin America, particularly in the Atlantic Forest ecoregion. Our data on survival, relative growth, and natural regeneration for various native tree species can help farmers select the most appropriate agroforestry species for the cultivation of *I. paraguariensis*.

Here, *C. fissilis* has shown susceptibility to disease and attack by leafcutter ants, so we suggest planting this species only after populations of such ants have been reduced and populations of their natural enemies have increased in agroforestry plots, which may be promoted by planting other tree species, such as *C. americana*, because of its high survival rate, *C. trichotoma*, because of its rapid growth, stem uniformity, and resistance to herbivorous insects, and *H. albus*, because of its early flowering, which accelerates natural regeneration.

In this work, agronomic systems and plantation types influenced the survival, growth, and natural regeneration of the six species in different ways. Given this variability, we recommend carefully considering the local context of the plots of interest, as well as considering cattle browsing and trampling as factors that can influence early establishment of seedlings. Ultimately farmers must make the final decision about the use of plots and the type of plantation in agroforestry systems, and they should consider all the relevant factors that can affect growth, survival, and regeneration.

**Supplementary Materials:** The following supporting information can be downloaded at https://www.mdpi.com/article/10.3390/f13122045/s1: Figure S1. Plot design; Figure S2. Experiment design in Caaguazú. Arrangement in (a) path side and islet plots and (b) field edge plots; Figure S3. Experiment design in Itapúa. Arrangement in (a) plots in the types of plantation and (b) plants on the plot; Table S1. Type of management in each agronomic system at the two study sites; Table S2. Initial average height measurements (m) of individuals planted in different experimental plots (p); Table S3. Plausible models for the survival, growth, and regeneration of planted species in the two agroforestry experiments; Table S4. Descriptive statistics of the survival, growth, and natural regeneration of planted species at the two study locations, stratified by agronomic system and plantation type and Table S5. Initial physical and chemical properties of the soil in the experimental plots in Caaguazú (Year = 2010).

**Author Contributions:** Conceptualization and methodology, A.I.O., J.M.R.B. and L.C.; data collection, A.I.O.; formal analysis, A.I.O. and L.C.; validation, A.I.O., L.C. and J.M.R.B.; writing—original draft preparation, A.I.O.; writing—review and editing, L.C. and J.M.R.B.; supervision, J.M.R.B. All authors have read and agreed to the published version of the manuscript.

**Funding:** We thank the Carolina Foundation of Spain and the National University of Asunción for a doctoral scholarship for A.I. and the National Council of Science and Technology (CONACYT) of Paraguay for support within the framework of the Scientific and Technological Linkage Program (PVCT19). We are also grateful to FIRE Paraguay for linking the research with the institutional project "Restoration of Agricultural Spaces".

**Data Availability Statement:** Not applicable.

**Acknowledgments:** We are grateful to the landowners who voluntarily allowed us access to their properties, without which this study would not have been possible. Finally, we thank Alice Romero and Anibal León for fieldwork, and Ulises Riveros and Verónica Cruz-Alonso for statistical support.

**Conflicts of Interest:** The authors declare no conflict of interest.

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
