# Peer review of "Establishment and Natural Regeneration of Native Trees in Agroforestry Systems in the Paraguayan Atlantic Forest"

_forests, doi:10.3390/f13122045_

Round 1

Reviewer 1 Report

Title. The title is appropriate to the subject, informative, and concise. This answers three important questions: What? Where? How?

Abstract. The abstract is concise, provides a clear overview, includes essential facts for the paper, and concludes with a final point that places the work described in a broader context.

Keywords. These are enough for the topic.

Introduction. The introduction includes background to provide an appreciation for the context of the work presented and also states the rationale and problem that the researchers attempted to answer through their experiments.

Line 53 — What are the obstacles? Specify them.

Material and methods.  In this section, the authors describe the correct steps that followed during conducting their study, give precise details of the study design, and how they analyzed the data.

Line 119 — It’s better ‘Experiment Description'

Line 149 — Ilex paraguariensis in italics.

Results. This section was well written and shows all data with good descriptions. The results say about the objective that motivates the research, and the authors take a broad look at their findings and examine the work in the larger context of the field.

Discussion. The authors had to discuss the data with respect to how their data fit into what is currently known in the field.

Conclusion. This section included the major conclusions, which were briefly written.

Figures and Tables. Both sections have good information and are necessary for the manuscript, they depict the data nicely.

Tables 2, 3, and 4 — What is the header of the second column?

Reviewer 2 Report

The scale in Fig. 2 for all tree species (a, b, c, d, e) should be up to a value of 100 %, as by Cordia americana (f)

On line 326, the citation Harmer 2001 should be replaced by number 87 as recorded in the bibliography.

The abstract significantly exceeds the required number of 200 words.

At the end of the article, before the list of references, a text about Supplementary Materials and Author Contributions, the content of which must comply with the Instructions for Authors, should be added

Reviewer 3 Report

Dear Authors, 

Thank you so much for preparing an impressive paper.

Please find some comments and suggestions on the manuscript in the attachment. 
